# Linking Human Betaretrovirus with Autoimmunity and Liver Disease in Patients with Primary Biliary Cholangitis

**DOI:** 10.3390/v14091941

**Published:** 2022-08-31

**Authors:** Hussain Syed, Tara Penner, Andrew L. Mason

**Affiliations:** 1Department of Medicine, University of Alberta, Edmonton, AB T6G 2E1, Canada; 2Center of Excellence for Gastrointestinal Inflammation and Immunity Research, University of Alberta, Edmonton, AB T6G 2E1, Canada; 3Li Ka Shing Institute of Virology, University of Alberta, Edmonton, AB T6G 2E1, Canada

**Keywords:** biliary epithelial cells (BEC), Bradford Hill criteria, human betaretrovirus (HBRV), Koch’s postulates, mouse mammary tumor virus (MMTV), primary biliary cholangitis (PBC)

## Abstract

Primary biliary cholangitis (PBC) is an autoimmune liver disease characterized by the production of diagnostic antimitochondrial antibodies (AMA) reactive to the pyruvate dehydrogenase complex. A human betaretrovirus (HBRV) resembling mouse mammary tumor virus has been characterized in patients with PBC. However, linking the viral infection with the disease is not a straight-forward process because PBC is a complex multifactorial disease influenced by genetic, hormonal, autoimmune, environmental, and other factors. Currently, PBC is assumed to have an autoimmune etiology, but the evidence is lacking to support this conjecture. In this review, we describe different approaches connecting HBRV with PBC. Initially, we used co-cultivation of HBRV with biliary epithelial cells to trigger the PBC-specific phenotype with cell surface expression of cryptic mitochondrial autoantigens linked with antimitochondrial antibody expression. Subsequently, we have derived layers of proof to support the role of betaretrovirus infection in mouse models of autoimmune biliary disease with spontaneous AMA production and in patients with PBC. Using Hill’s criteria, we provide an overview of how betaretrovirus infection may trigger autoimmunity and propagate biliary disease. Ultimately, the demonstration that disease can be cured with antiviral therapy may sway the argument toward an infectious disease etiology in an analogous fashion that was used to link *H. pylori* with peptic ulcer disease.

## 1. Introduction

### 1.1. Primary Biliary Cholangitis

Primary biliary cholangitis (PBC) is a rare cholestatic liver disease characterized by immune damage to intrahepatic bile ducts that may progress to cirrhosis and liver failure [1,2]. PBC is classified as an autoimmune disease because most patients (80–95%) make antimitochondrial antibodies (AMA) [1,2]. It is often assumed that PBC is caused by the autoimmune response, but the etiology of the disease is unknown [3]. PBC is diagnosed when patients present with elevated alkaline phosphatase, a positive serum AMA, or by liver biopsy in AMA-negative patients [1,2]. The target of AMA is pyruvate dehydrogenase E2 protein (PDC-E2) that is overexpressed on the cell surface of biliary epithelium cells (BEC) and in perihepatic lymph nodes [4,5]. It is thought that the aberrant expression of PDC-E2 then leads to the production of AMA and autoimmunity [4,5]. Genome-wide association studies have identified genes linked with PBC, and epidemiological studies strongly suggest that environmental factors may trigger disease in genetically susceptible individuals [6]. Bacteria or exposure to xenobiotics have been proposed as external triggers, and we have focused on the role of human betaretrovirus (HBRV) infection [7].

### 1.2. PBC: Epidemiology and Pathophysiology

Similar to other autoimmune disorders, PBC is ~10 times more common in women [2]. Hormone replacement therapy and a younger age of first pregnancy both provide an increased risk of developing PBC [1,2]. PBC is a rare disease found in all parts of the world, with a prevalence ranging from 1:2500 to 1:100,000. There is an increased prevalence moving away from the equator, with geographical clustering in areas of North America and Europe. Indigenous Canadians have a markedly elevated prevalence of PBC and a worse prognosis compared with Canadians of European descent [8,9,10]. We will return to the model of how this may be linked with an increased burden of genetic predisposition and exposure to European-derived pathogens [11].

Histologically, patients develop non-suppurative cholangitis with the immune destruction of interlobular bile ducts. The progressive ductopenia leads to bile accumulation, resulting in fibrosis and cirrhosis. Therefore, choleretics are the mainstay of treatment, but they are not curative. In fact, patients unresponsive to the standard of care account for 5% of patients awaiting liver transplantation in North America [12]. PBC patients suffer from fatigue comparable with chronic fatigue syndrome/myalgic encephalomyelitis [13,14]. Some of the exhaustion has a peripheral etiology, as PBC patients experience prolonged muscular acidosis with a slower rate of recovery of phosphocreatine levels following exercise [13,15]. The cause of fatigue is unknown but may persist in up to 50% of patients following liver transplantation despite having no apparent liver disease [16,17].

### 1.3. PBC: Genes vs. Environment

Genetic factors are involved in the development of PBC. The disease occurs more frequently in monozygotic versus dizygotic twins and is more common in related family members [6]. Genes involved in innate and adaptive immunity have been linked with PBC, but not everyone with a genetic predisposition will go on to develop PBC, suggesting that any environmental agent only causes disease on a specific genetic background [6]. Genetic studies have linked PBC with complement deficiency [18,19], specific HLA class II alleles, the IL-12 and interferon (IFN)-γ cytokine axis, and other innate immune genes [6,20,21,22,23]. As commonly observed in various idiopathic autoimmune diseases, the genetic predisposition in PBC is polygenic in nature, but the effects of specific variants are poorly understood [6]. 

More recently, specific PBC populations have been found to have diminished IL-12A production [24], and a clinical trial using the IL-12 inhibitor ustekinumab showed no efficacy [25], suggesting that patients may lack IL-12 activity. Epidemiological studies also indicate that PBC patients may lack immune function with both an increased prevalence of infectious diseases, such as tonsillitis and urinary tract infection, and an increased prevalence of cancers, such as lymphoma, thyroid, kidney, and breast cancer [7,26,27].

The geographical clustering suggests the involvement of environmental factors or infectious agents. For example, unrelated family members and caregivers may develop PBC, suggesting a transmissible factor. Furthermore, children moving from low prevalence to high prevalence countries have an increased risk of developing PBC [7]. The recurrence of PBC in up to 50% of patients following liver transplantation is suggestive of an infectious process [28]. Of note, several bacteria, xenobiotics, and viruses have been implicated in the etiology of PBC, but no specific agent has been directly linked with the disease [29,30,31,32,33].

The role of bacteria triggering PBC through the mechanism of molecular mimicry has been extensively studied, but none of the candidate bacterial agents have been confirmed in case-control studies [29,34]. The link between bacterial urinary tract infection and triggering PBC has been hypothesized but never demonstrated, and no candidate bacteria have been localized in PBC bile ducts. Recent bacterial studies in PBC have focused on the dysbiosis in the gut microbiome and the effect of bile salts in modulating the disease process [34], whereas no original articles on mono-bacterial infection in PBC have been reported for over 10 years. The role of xenobiotics in PBC is supported by clustering around toxic waste sites [35] and animal models, where xenobiotic immunization resulted in AMA production and cholangitis, but no specific agent has been linked with PBC [36,37].

### 1.4. PBC: Role of Autoimmunity

PBC is an autoimmune disease because patients make humoral and cellular autoimmune responses to the lipoylated domains PDC-E2 and related mitochondrial oxo-acid dehydrogenases [38,39,40,41]. Because of the lack of a defined environmental trigger and polygenic predisposition of unknown significance, it is often presumed that PBC also has an autoimmune etiology. However, there is insufficient evidence to support this assumption. 

The Witebsky criteria for establishing an autoimmune etiology have not been met. For example, PBC has not been recreated in animal models by the infusion of AMA or autoreactive T cells; vaccination with PDC-E2 and an adjuvant generates AMA but does not cause biliary pathology [42,43].Humoral autoimmunity does not propagate disease because AMA are not necessary to acquire PBC. AMA-negative patients develop the same disease process as AMA-positive individuals, and some AMA-positive individuals do not develop PBC [44]. No other autoantibodies are necessary to generate PBC.With regard to cellular immunity, PBC patients appear to have an insufficient precursor frequency of PDC-E2 reactive T lymphocytes to generate disease. The autoreactive T cells are estimated to be approximately 1 × 10^−5^ in the liver and 1 × 10^−7^ in the blood [45]. In comparison, preliminary studies suggest the frequency of HBRV reactive lymphocytes is approximately 100-fold higher in PBC patients with HBRV infection [46]. The level of anti-HBRV cellular immune responses is comparable with observations from other infectious diseases, such as chronic hepatitis C virus infection [47,48].While immunosuppression is the mainstay of treatment for autoimmune hepatitis, this therapeutic approach is of limited utility for PBC patients [49,50]. Moreover, liver transplant recipients are more likely to develop earlier and more severe recurrent PBC disease with the use of more potent immunosuppression regimens [51].Much of the assumption that PBC is an autoimmune disease is based on mouse models with autoimmune biliary disease that develop spontaneous AMA production [52]. Notably, these mouse models also have evidence of betaretrovirus infection with mouse mammary tumor virus (MMTV) protein expression [53]. We will address this process in Section 2.4, as betaretrovirus infection is directly linked with the production of AMA [7,54].Notably, the NOD.c3c4 mouse has evidence of MMTV in bile ducts, and intervention with combination antiretroviral therapy (ART) has been shown to ameliorate the cholangitis in this PBC mouse model [53,55].In case series and clinical trials using combination ART to treat PBC patients, improvements in cholangitis have been linked with reduced viral load [56,57,58,59,60]. Notably, the gold standard of meeting randomized controlled trial study endpoints has not been achieved to date due to the lack of tolerability of combination ART and the lack of antiviral efficacy of repurposed ART used to treat HIV.

In summary, the relative role that autoimmunity plays in initiating or propagating the disease process is unknown but an important question to resolve [7]. It is increasingly recognized that autoimmune diseases in general, such as inflammatory bowel disease and PBC, are linked with dysregulated immunity, which is in part related to the genetic architecture that predisposes them to disease [6,61]. For example, PBC patients have an increased susceptibility to infectious diseases as well as an increased prevalence of several cancers, in keeping with diminished immune surveillance [26,27]. Moreover, multiple reports document diminished immunity in PBC with low complement C4 levels, decreased levels of B and T cell subsets, as well as reduced parameters indicative of lymphocyte activity in general [62,63,64,65]. Consequently, the question of whether PBC has an autoimmune or an infectious etiology is important to resolve because it has a direct bearing on the translational investigation of the disease process. On the one hand, we have clinical trials emphasizing immunosuppression [25] and, on the other, trials adopting an antimicrobial approach [56,57,58,59,60,66,67] (Figure 1).

## 2. Human Betaretrovirus (HBRV)

### 2.1. Betaretroviruses

The human betaretrovirus (HBRV) is the only exogenous betaretrovirus characterized in humans to date [68]. HBRV is not endogenously encoded in the human genome [68,69,70] but has been mistakenly been referred to as a human endogenous retrovirus by others [71]. HBRV shares very close nucleotide similarity with MMTV, to such an extent that human and mouse isolates are not easily distinguishable [54,68,70]. Notably, HBRV infection has been detected in humans dating back to the copper age [72] and the genomic similarities with MMTV strongly suggest a zoonotic transmission, which may have first occurred with the development of agriculture and storage of grain [73,74]. The relatively conserved genome sequences of MMTV and HBRV, and the limited variation between different isolates in general, may be attributable to the increased processivity of the betaretrovirus polymerase [75]. The increased rate of DNA synthesis facilitated by MMTV RT helps to counteract the deaminase activity and extent of G-to-A mutations mediated by the APOBEC3 restriction factor, thus limiting the development of genomic variants [76].

### 2.2. Mouse Mammary Tumor Virus (MMTV)

MMTV is an established cause for breast cancer, kidney cancer, and lymphoma in mice and has also been linked with inflammatory and autoimmune disorders. MMTV infection is endemic in mice, where infection can be passed endogenously in the genome and exogenously in breast milk [77]. MMTV viral burden is highest in lactating breast and virus production rises with parity; the viral levels are lower in lymphoid tissue, and very low viral levels are detected within the liver (biliary epithelium and lymphocytes) and salivary glands [77,78,79,80]. Of note, a similar tissue distribution of HBRV has been observed in humans [54,69,81]. 

Weanling mice are exposed to MMTV in breast milk before acidification occurs in the stomach, but infection can be transmitted by biting as well. When neonatal infection occurs through the gastrointestinal tract, mice become tolerized to MMTV and fail to make adequate neutralizing antibody responses [78,82]. MMTV replicates in dividing lymphocytes within the gut-associated lymphoid tissue cells and requires a virally encoded superantigen to stimulate lymphocyte proliferation. Because the viral load is below the limits of detection in blood and antibody production is suboptimal, superantigen induced proliferation and then deletion of T cell receptor Vβ subsets is used to diagnose MMTV infection in mice [78,83].

### 2.3. HBRV and Human Disorders

HBRV (also referred to as the human mammary tumor virus) was first detected in patients with breast cancer over 50 years ago, in lymphoma and other female hormone responsive cancers, and subsequently in patients with PBC [73,81,84,85,86,87,88]. HBRV has been isolated from patient’s lymph node samples and passaged to primary epithelial cells [68]. Gold standard evidence of infection in humans has been derived by the demonstration of multiple proviral integrations in patients’ tissue samples [69]. It is not known how HBRV is transmitted, but the agent has been detected in saliva and milk, similar to observations in mice [72,73,81,84,86,87,88,89].

In 2003, our group characterized HBRV in patients with PBC and established a link between viral infection and autoimmunity [54]. In the first instance, we demonstrated evidence of infection in BEC and in perihepatic lymph node tissues that also overexpressed the PDC-E2 mitochondrial autoantigens (Figure 2A). Then, in co-cultivation studies, we showed that the virus could stimulate the PBC phenotype of increased and aberrant expression of PDC-E2 in healthy BEC (Figure 2B). Indeed, the expression of cryptic mitochondrial antigens is thought to be directly related to the generation of AMA [4], and the co-culture studies directly link viral infection with autoimmunity.

In subsequent clinical studies, HBRV has been more directly linked with cholangitis. Using ligation-mediated PCR and next generation sequencing, we identified HBRV proviral integrations in the biliary epithelium of the majority of PBC patients tested [69] (see NCI Retroviral Integration Database; rid.ncifcrf.gov [90]). In several case series and clinical trials, ART has been shown to improve histological and biochemical outcomes, and reduce viral load with combination ART [56,57,58,59,60,66,67]. Both humoral and cellular immune responses to HBRV have been demonstrated in patients with PBC [46,91].

### 2.4. Challenges Linking HBRV with PBC

The interaction of genetic and external factors in idiopathic disorders can be difficult to study, especially when those diseases occur at very low rates. PBC is found with a prevalence of ~1:3000 in the general population [1,2], whereas HBRV appears to be a relatively common infection in patients with liver disease [69,92]. Accordingly, we have been studying the hypothesis that HBRV may only trigger PBC in patients with genetic susceptibility [7]. Because HBRV may be found in biliary epithelium and intrahepatic lymphocytes, viral infection may also contribute to the development of liver disease in other circumstances, such as in patients with alcohol-related hepatitis [92]. While a worse prognosis has been observed in patients with alcohol-related liver disease and viral co-infection [93], this has not yet been assessed in patients with HBRV.

A second challenge encountered with studying the role of HBRV in PBC includes the difficulty of detecting the low viral burden in patients, the lack of antibody production, and a lack of diagnostic assays to diagnose HBRV in general [7,69,91]. Indeed, the demonstration of superantigen activity has been the standard for detecting MMTV infection in mice because PCR cannot distinguish genomic viral RNA from endogenously-expressed MMTV transcripts, and mice infected by breast milk have suboptimal neutralizing antibody responses [78,82]. In fact, it is likely that the lack of reproducible diagnostic assays led to the retroviral community’s lack of interest in pursuing the study of HBRV pathogenesis in breast cancer [87] and, more recently, in patients with PBC [7] because, without such assays, the debates became polarized, without clear resolution.

### 2.5. Linking HBRV with AMA Production

Several mechanisms have been proposed for how environmental agents may trigger autoantibody production [29]. Molecular mimicry of microbial and host proteins is the most prevalent explanation for the generation of AMA as a result of bacterial PDC-E2 proteins resembling mitochondrial autoantigens [94]. Indeed, several candidate bacteria have been proposed as causative agents in propagating PBC by the process of molecular mimicry, but none have been linked nor confirmed with the development of cholangitis [29,95]. A recurrent issue with the microbial molecular mimicry hypothesis for generating autoimmune diseases, in general, is that the mechanism is often presumed as a cause but not proven, except as a side effect from vaccination [29]. Furthermore, it is unclear how a remote bacterial infection distant from the biliary epithelium in PBC patients can promote biliary apoptosis and PDC-E2 expression in cholangiocytes or attract IgA immune complexes carrying bacterial molecular mimics into the epithelium cells to trigger the autoimmune destruction of cholangiocytes (Figure 3A).

Homology searches of the betaretrovirus proteins show no resemblance to mitochondrial antigens or other autoantigens associated with PBC to support the concept that molecular mimicry promotes autoimmunity. Rather, we have suggested a different model for the generation of autoimmune responses to PDC-E2 as a result of betaretrovirus infection (Figure 3B). As shown in Figure 2B, HBRV triggers the aberrant expression of PDC-E2 within infected cholangiocytes. This, in turn, leads to the expression of cryptic mitochondrial antigens on the cell surface, and because retroviruses incorporate host proteins into the viral envelope when egressing from the cell surface, we propose that PDC-E2 is taken up into the virion particle (Figure 3B) [29]. Then viral and host proteins are presented by antigen-presenting cells (APC) to lymphocytes, triggering AMA production and autoreactivity to PDC-E2 (Figure 3B). Accordingly, we hypothesize that autoimmunity is a by-product of betaretroviral infection. 

### 2.6. Linking MMTV with AMA Production in Mice

The same process has been documented in multiple mouse models that develop spontaneous AMA production linked with MMTV protein expression and immune disruption [53]. In these models, there is little association with cholangitis per se. Rather, mice have diminished immunity from genetic manipulation that likely permits the expression of endogenous MMTV and the development of inflammatory disease. Indeed, some investigators have questioned whether the AMA-producing mice present adequate models of PBC because few have evidence of cholangitis [96]. Certainly, the NOD.c3c4 mouse develops cholangitis related to MMTV in bile ducts [97,98], whereas the other AMA-producing mice, such as the IL-2 Receptor α deficient mouse [99], the T cell TGF-β receptor II dominant-negative [100], and the Scurfy mouse lacking T regulatory cells [101] die from diffuse inflammatory disease. Nevertheless, the NOD.c3c4 mouse model of PBC has proven useful in demonstrating that MMTV is related to liver disease. The NOD.c3c4 model “spontaneously” expresses AMA, has evidence of PDC-E2 expression and MMTV protein expression in bile ducts and lymphoid tissues, and develops cholangitis [53].

### 2.7. Linking MMTV with Cholangitis in Mice 

To assess the role of MMTV in the development of cholangitis, NOD.c3c4 mice were treated with repurposed HIV reverse transcriptase (RT) inhibitors and HIV protease inhibitors with a known antiviral activity to MMTV [102]. The improvement in biochemical and histological cholangitis occurred, with all regimens coinciding with a reduction in hepatic MMTV RNA, thus limiting the concern for potential off-target effects of ART modulating inflammatory responses [55]. A biological gradient was observed with the more potent combination of ART regimens incorporating a protease inhibitor, resulting in improved biochemical and histological responses versus treatment with RT inhibitors alone. Indeed, mice on lamivudine and zidovudine experienced antiviral resistance with YMDD mutations and biochemical breakthroughs due to insufficient antiviral therapy. 

Histological cholangitis was only readily apparent in NOD.c3c4, whereas the other AMA-producing mouse models still displayed lymphoid expression of aberrant PDC-E2 and MMTV proteins [53]. The models with diffuse inflammatory disease, which lack cholangitis, more clearly link MMTV with the production of AMA. However, MMTV expression is common in immune-deficient mice, and because the genetically modified mice that have immune deficiency produce AMA without biliary disease, it is not surprising that other researchers have questioned whether murine AMA is adequate to model autoimmune biliary disease in mice [96], whereas AMA is one of the diagnostic criteria for patients with PBC [1,2].

It is not clear why the NOD.c3c4 model develops viral cholangitis as compared with the other MMTV-expressing models with AMA production. However, it may be related to the biliary cyst formation observed in the NOD.c3c4 line. This phenotype is not observed in patients with PBC but rather in subjects with polycystic liver and kidney disease. Further studies in NOD.c3c4 reveal that the proliferating biliary epithelium is related to a mutated *pkhd1* gene encoding fibrocystin [103]. The impairment of fibrocystin alters the sensing function of cholangiocyte cilia, which causes cholangiocyte hyperproliferation with biliary cyst formation. Because MMTV requires dividing cells for the efficient delivery of the preintegration complex into the nuclei [104], it is likely that the proliferating cholangiocytes are sufficiently permissive for MMTV to augment virus production in the NOD.c3c4 mouse. Accordingly, the NOD.c3c4 provides an example of combined genetic predisposition and betaretrovirus expression to generate autoimmunity and cholangitis, whereas the other AMA-producing models demonstrate AMA production to be a result of immunodeficiency, leading to MMTV expression without predominant cholangitis [53,55].

### 2.8. Potential Mechanisms for Betaretrovirus-Induced AMA Production

It unknown how betaretrovirus infection modulates PDC-E2 expression, but studies point towards viral induction of anabolic metabolism. As part of the viral pathogenesis of mouse breast cancer, MMTV initiates cellular proliferation by insertional mutagenesis proximal to *Wnt1* and *Fgf3* genes to activate the Wnt/beta-catenin signaling pathway [78,105]. This leads to the induction of glycolysis, anabolic metabolism and the biosynthesis of nutrients required for cellular proliferation [105]. By studying PBC cholangiocytes with increased PDC-E2 expression, we have observed a similar metabolic remodeling linked with an upregulation in glycolysis, inhibition of oxidative phosphorylation, and compensatory mitochondrial biogenesis leading to accumulation of PDC-E2 [7,106,107]. Similar metabolic remodeling can be induced in human epithelial cells in vitro by over-expressing both WNT1 and FGF3 to induce glycolysis and mitochondrial biogenesis [105].

Apart from the metabolic changes, other viral mechanisms are likely involved in generating autoimmune responses in PBC patients. One such mechanism likely involves the viral superantigen required for viral replication [29,78]. The betaretrovirus superantigen binds MHC II and the T cell receptor outside of the peptide binding grove to amplify lymphocyte proliferation in a TCR-Vβ restricted manner. It has been suggested that this process may induce proliferation of autoreactive B and T lymphocytes [29]. However, the superantigen may also impact antigen presentation as well. Specifically, MMTV superantigen has been shown to displace the class II invariant chain peptide required for stabilizing MHC class II [108]. This occurs in the endoplasmic reticulum prior to antigen peptide loading and therefore may subsequently influence the nature of viral peptide versus autoantigen presentation. Additional studies are required to better understand how betaretroviruses induce autoimmunity. 

## 3. Modified Koch’s Postulates Linking HBRV with PBC In Vitro

Koch’s postulates were established in the 19th century and originally intended to link acute infections with a clear host–pathogen relationship. As viruses are more difficult to isolate, Fredericks and Relman proposed updating the criteria to include molecular detection methods for the detection and verification of infection [109]. However, chronic viral infections pose a particular challenge, as not all hosts infected with a virus will exhibit disease symptoms. This is especially the case in chronic autoimmune diseases such as PBC, where genetic predisposition modulates the response to infection and disease progression. 

We have suggested that the modified Koch’s postulates in vitro may serve as a framework for linking HBRV with the autoimmune phenotype of PBC [110]. 

*Agent must be found in all patients with disease and not healthy control subjects.* HBRV has been found in the majority of PBC patients using different methods, but also in patients and healthy subjects without PBC (Table 1; *Specificity*) [54,69,91,92].*Agent must be isolated from patients with disease*; HBRV has been isolated using Hs587T in co-culture with lymph node homogenates from PBC patients [44].*Agent must cause disease when introduced to a healthy organism*. HBRV and pure MMTV isolates induce the autoimmune phenotype in healthy BEC in vitro [54].*Agent must be re-isolated from the infected source*. HBRV was isolated from conditioned media and used in serial passage to induce the autoimmune phenotype [54] (Figure 2B).

The validity of this co-culture model depends on several factors that include the specificity of the PBC autoimmune phenotype, whether confounding factors may trigger the same appearance, and the reproducibility of the model. The production of AMA is firmly linked with PBC but is also detected in a proportion of patients with autoimmune hepatitis and rheumatologic diseases, such as systemic sclerosis, Sjögren’s syndrome, and systemic lupus erythematosus [111]. However, the autoimmune phenotype of aberrant PDC-E2 expression in humans has only been demonstrated in patients with PBC [4] but is probably not extensively sought in other diseases. Selmi and colleagues questioned whether aberrant expression of PDC-E2 may be caused by the induction of apoptosis in cultured BEC rather than HBRV [112]. However, no cell death was observed in our HBRV co-culture studies [54]. Furthermore, when apoptosis is observed in biliary epithelium displaying the aberrant PDC-E2 in PBC patient liver biopsies, it is exclusively observed with lymphocyte infiltrate and immune destruction of bile ducts [113]. 

Questions were also raised about whether the immunochemistry was non-specific and caused by artefacts from the dilution of antibodies in the colocalization immunohistochemistry studies [112]. However, this seems unlikely, as the presence of HBRV was detected in lymph nodes using betaretrovirus anti-P27 Capsid and anti-gp52 Surface antibodies (Figure 4), and aberrant PDC-E2 expression was detected using two different antibodies as well with PBC patient-derived AMA and the c150 murine monoclonal AMA [54,71]. As to the question of reproducibility of the co-culture studies, the same findings have now been demonstrated in three separate labs [54,114].

While Koch’s postulates cannot ultimately prove an infectious cause of a complex disease, exploring the evidence supporting each criterion can strengthen the association into a clear cause-and-effect relationship. For multipart biological relationships, it is not possible to prove an association without the ability to control each and every variable linked to a specific disorder. Consequently, criteria such as these may serve as frameworks rather than ultimate benchmarks because of the complexity of the disease [109,110,115].

## 4. Bradford Hill Causal Criteria for Connecting HBRV and PBC

Another strategy for determining whether an agent causes a disease is the use of Bradford Hill causal criteria, which provide evidence for an association between a factor and disease, eventually leading to the determination of causation [115]. A condensed version of Hill’s criteria linking PBC with HBRV is tabulated below (Table 1). In this Special Issue of “Human Betaretrovirus and Related Diseases,” Lawson and Glenn use Hill’s criteria to illustrate a plausible and comprehensive relationship between HBRV and breast cancer [89]. Originally published in 1965, Hill’s criteria have been modified by various authors as we better understand the biological mechanisms of disease [116]. Factors such as genomics, molecular toxicology, and data integration have been incorporated into the original criteria to broaden the scope of what is considered causation. 

### 4.1. Strength of the Association between HBRV and PBC

The strength of the association refers to the link between exposure to the factor in question and the disease. Despite the challenges outlined in Section 2.4, numerous lines of evidence have been amassed to show that HBRV plays a role in PBC. 

The first evidence suggestive of viral infection and the development of PBC was the detection of cross-reactive antibodies to known retroviruses in the serum of PBC patients [117]. However, the seroreactivity was non-specific and also found in patients with other cholestatic disorders, including primary sclerosing cholangitis and biliary atresia. Subsequent ELISA studies using HBRV gp52 Surface protein and incorporating large populations of age-matched healthy subjects and blood donors showed that HBRV reactivity was more specific for patients with PBC and breast cancer [91].

**Table 1 viruses-14-01941-t001:** Hill’s criteria applied to HBRV infection and PBC.

Criteria	Link of HBRV Infection with PBC	References
*Strength of association **	HBRV integrations in the majority of PBC patients’ bile ducts ○ Intrahepatic lymphocytes tested to date Betaretroviruses induce mitochondrial autoantigen expression in vitro ○ MMTV in biliary epithelium and lymphoid tissue in PBC models ○ HBRV in PBC biliary epithelium and lymphoid tissue Betaretroviruses are linked with cholangitis ○ NOD.c3c4 mouse models Rx combination antiretroviral therapy ○ PBC patients respond to combination ART	[69] [46] [54,114] [53,55] [54,70] [55] [55] [56,57,58,59,60] [56,57,58,59,60,66,67]
*Specificity **	PBC patients only develop disease with a specific genetic background Lack of specificity for HBRV in PBC; HBRV may be a hepatotropic virus: ○ HBRV detected in autoimmune liver disease: PBC, autoimmune hepatitis ○ Idiopathic liver disease: cryptogenic cirrhosis, HCC ○ Alcoholic-associated liver disease: HBRV acts as a cofactor to augment disease AMA production is linked with HBRV/MMTV and highly specific for PBC AMA is not specific to autoimmune biliary disease in mice but linked with MMTV	[6,21] [69,92] [69] [68,69,92] [92] [1,2,53,54] [53,96] [53]
*Consistency **	HBRV has been detected in tissues and by immune response in PBC patients ○ Biliary epithelium: Proviral integrations (58%), electron microscopy, RT-PCR ○ Lymph nodes: Proviral integrations (45%), RT-PCR, immunochemistry (75%) ○ Liver: Proviral integrations (13%), Nested PCR (12%), RT-PCR (20%) ○ Immune responses: Flow cytometry CD8+ PBMC (43%), interferon-γ release assay in PBMC (50%) and intrahepatic lymphocytes (100%), IgG reactivity to HBRV Su (11%)	[54,69] [54,69] [54,69,92] [46,91,118] [46,91]
*Biological gradient*	Combination vs single ART ↑ resolution of cholangitis in mice and humans ↑ potent immunosuppression leads to earlier and more severe recurrence after liver transplant ○ In keeping with an infectious disease process versus an autoimmune response	[55,56,57,58,59,60,66,67] [28,51]
*Temporality*	Cholestasis soon after liver transplantation predicts PBC recurrence > 50% patients ○ ↑ alkaline phosphatase or bilirubin in the first year following liver transplantation doubles risk	[28,51]
*Plausibility Coherence **	Female adults develop PBC exacerbated by hormone replacement therapy: ○ Female hormone response elements in HBRV LTR drive viral replication Betaretroviruses ↑ mitochondrial protein expression by inhibiting mitochondria ○ Mitochondrial inhibition may cause chronic fatigue associated with PBC Combination ART improves symptoms, hepatic biochemistry, and histological cholangitis ○ Immunosuppression and choleretic therapy do not impact symptoms	[26] [119] [54,106,107] [1,2] [56,57,58,59,60] [1,2]
*Analogy*	Numerous viruses can cause cancer, inflammatory diseases, and autoantibody production modulated by age, sex, race, and genetic predisposition. ○ Female patients with hepatitis C virus infection and the AH8.1 ancestral haplotype more commonly develop autoantibodies and vasculitis. ○ Other viruses linked with breast cancer: Epstein Barr virus and papilloma virus	[120,121,122] [6,123] [124]

* Several criteria may overlap in function.

Additional electron microscopy studies identified B-type betaretrovirus-like particles in biliary epithelium extracted from PBC patients receiving liver transplantation and also in conditioned media from PBC lymph node co-culture studies [54]. These particles were observed in all three patients with PBC, but only one particle was detected in five control patients’ biliary epithelium. Conditioned media from the co-cultivation studies also contained virus-like particles with the hydrodynamic and genomic features of HBRV and the morphological appearance of B-type particles with an electron-dense nucleocapsid adjacent to the envelope (Figure 5). Subsequently, transmissible HBRV virions have been isolated from PBC lymph nodes that display the appearance of B-type particles with an asymmetric nucleocapsid similar to MMTV, as recently reported in this Special Issue [68].

The question has arisen of why viral particles have not been visualized in electron microscopy studies of PBC samples by other groups [112,125,126]. Without foreknowledge of what other investigators were seeking, it is not possible to provide a clear answer. Our studies, however, focused on assessing the presence of virus particles in biliary epithelial cells extracted from liver transplant recipients [54]. Because these primary cells were proliferating in the absence of immune surveillance, it is possible that the cells could support viral replication, as observed in the co-culture experiments (Figure 2B).

The HBRV genome was first detected in a PBC biliary epithelium cDNA library using degenerate PCR primers capable of amplifying any retroviral *pol* sequence, and the sequence was very closely related to MMTV with 97–98% identity [54]. Subsequently, the proviral genome was cloned and sequenced from lymph node DNA [70], and in a recent study published in this issue, HBRV was isolated from PBC lymph nodes and sequenced [68]. The human- and mouse-derived betaretrovirus nucleotide sequences share near identity but display consistent differences expected from isolates derived from separate species [68].Initial RT-PCR studies to detect HBRV RNA in the liver and perihepatic lymph node samples revealed that lymph nodes had the highest viral burden [54]. Viral proteins were easily detected by immunochemistry in PBC lymph node samples (Figure 2 and Figure 4) but not liver, where HBRV RNA was found in 73% of perihepatic lymph nodes from PBC patients and also in 20% of control patients. Liver tissue had considerably less viral burden, where 29% of the livers from PBC patients were positive for viral cDNA, compared with 7% of control patient livers [54].

While other groups have not been able to replicate these results, it is important to note that the reports are compatible with each other, considering the studies were restricted to the liver rather than lymphoid tissue and the different viral detection methods employed. Notably, no other laboratory studied perihepatic lymph nodes for evidence of HBRV infection, where the highest viral burden was detected in our lab; we found evidence of viral infection in approximately 75% of PBC patients’ lymph nodes expressing HBRV RNA and proteins (Figure 4) [54]. It is conceivable that HBRV is likely below the limits of detection in the liver by PCR, as the agent exists with a very low viral burden in mice and humans. Both Selmi and colleagues and our lab reported no evidence of HBRV DNA in the whole liver when using a single round of PCR [54,112]. Indeed, nested PCR only identified HBRV in 20% of the liver samples [54], as confirmed by Johal and colleagues [92]. In the electron microscopy studies, we observed viral particles in 1:200 biliary epithelial cells from patients with PBC. As biliary epithelium cells only comprise 2–3% of the liver, it is possible that viral infection is below the limits of detection for PCR with one virus per 10,000 liver cells. In subsequent proviral integration studies that employed both ligation-mediated (LM) PCR and next-generation sequencing, we identified viral infection in biliary epithelium ex vivo in the majority of patients studied and confirmed that the viral integrations had a much higher prevalence in perihepatic lymph nodes than in the liver [69]. However, the methods used to detect HBRV insertions are cumbersome, expensive, and not easily adapted for general lab use.

Selmi and colleagues suggested “In our opinion, the only possible final evidence for a role of a betaretrovirus in PBC could be provided by the direct demonstration of the insertion of viral sequences in the genome of a large number of patients with PBC” [112]. As the gold standard for confirming retroviral infection is the detection of proviral integrations in the genome, we used ligation-mediated PCR to identify insertions of HBRV in the human genome [69]. We also employed next-generation sequencing to improve the sensitivity of detection and screened the sequences using a stringent pipeline to avoid picking up endogenous retroviruses, false positives from contaminated laboratory reagents, duplicates, or other problematic sequences. After screening, 58% of PBC patients were found to have HBRV integrations in their cholangiocytes, compared with only 7% of control patients. HBRV RNA was detected in the biliary epithelium of 58% of the PBC patients and in 15% of the control patients using a QuantiGene hybridization method, and 75% of the PBC patients versus 13% of the control cholangiocytes using in situ hybridization. Taken together, these results provide robust evidence for infection with an exogenous HBRV virus in PBC patients, with a higher number of retroviral integrations in BEC compared with the liver [69].Another line of evidence for the involvement of HBRV in PBC comes from serological studies. Using MMTV isolates, Selmi and colleagues were unable to detect any serological reactivity in PBC patients’ serum [112]. To create a more suitable assay, we expressed HBRV gp52 surface protein in human cells and used the soluble HBRV gp52 to create an ELISA assay. We observed HBRV gp52 antibody response in sera from 11.5% of the PBC patients, which was similar to that detected in breast cancer patients (10.2%); blood donors and age-matched control subjects had a 2–3% seroprevalence [91]. In total, these studies suggest that HBRV may be quite prevalent in the general population.As the proportion of PBC patients with viral integrations exceeded the seroprevalence of anti-HBRV gp52 surface reactivity, we performed further studies to determine whether HBRV may inhibit immune responses. Retroviruses may harbor immunosuppressive domains in their envelope protein that inhibit immune responses by stimulating the regulatory cytokine, interleukin 10 [127]. By immunoscreening healthy peripheral blood mononuclear cells with overlapping 15–20mer peptides derived from the HBRV envelope protein, we characterized three immunosuppressive domains that trigger IL-10 to inhibit immune responses [128]. Notably, a similar process has been described in neonatal mice, where MMTV infection triggers IL-10 production and inhibits the production of neutralizing antibodies [82].To study the cellular immune responses to HBRV infection in patients with liver disease and breast cancer, we have stimulated lymphocytes with overlapping Gag and envelope peptides and used flow cytometry, ELISpot, and ELISA to measure interferon-γ release. Our preliminary studies suggest that 100% of the intrahepatic lymphocytes and 50% of the peripheral blood mononuclear cells from PBC patients make interferon-γ as compared with ~10% of liver disease controls [46].

Taken together, these data support the hypothesis that PBC patients have evidence of betaretrovirus infection in the biliary epithelium and perihepatic lymph nodes and make both humoral and cellular immune responses to HBRV. 

### 4.2. Specificity of the Association between HBRV and PBC

HBRV infection is not specifically associated with PBC. In fact, Hill’s original meaning for specificity was that an exposure or association specifically causes one disease, and even though many diseases have multiple factors, Hill stated: “If we knew all the answers we might get back to a single factor” [115]. As we learn more, specific mechanisms can be proposed with specific relationships, and complex effects can be distilled to the most likely cause. 

Because PBC is a rare disease, HBRV infection is likely more prevalent than PBC. This clearly points towards the influence of other factors, such as sex, female hormones, increasing age, genetic predisposition, and other unknown environmental factors that all play a role in the development of autoimmune liver disease. 

Our studies suggest that HBRV is not specific for PBC, because the virus can be detected in up to 3% of healthy subjects, 35–40% of patients with breast cancer [46,87,89], and a proportion of patients with cryptogenic liver disease [69,92]. Johal and colleagues found evidence of HBRV in patients with chronic hepatitis C or B infection (21%), alcoholic liver disease (47%), non-alcoholic fatty liver disease (32%), and other types of liver disease (30%) using nested PCR [92]. While the latter technique may overestimate the proportion of patients with infection, our integration study also identified infection in 17% of patients with cryptogenic liver disease and 50% of autoimmune hepatitis (AIH) patients [69]. From these data, we can deduce that either HBRV is a passenger virus without any clinical significance or posit that HBRV is a hepatotropic virus and infection may contribute to other types of liver disease in addition to PBC. Notably, when hepatitis C virus was first identified, the virus was found in a wide range of liver disease patients with alcoholic liver disease, hepatitis B coinfection, AIH, hepatocellular carcinoma, and cryptogenic cirrhosis [69].

Because HBRV infection is found in liver disease patients without PBC and in breast cancer patients without liver disease, the most parsimonious hypothesis is that only the combined effects of HBRV and genetic predisposition provide the specific risk for developing PBC or AIH [110]. However, complex studies using a combination of genome risk score analysis to evaluate the PBC genetic predisposition will need to be conducted in conjunction with studies to evaluate HBRV infection to address this specific hypothesis. The main barriers at present are the lack of reproducible blood tests to detect HBRV as well as a lack of a validated genome risk score to perform the genetic studies. Based on the integration studies and preliminary cellular immune assay detecting HBRV reactivity in intrahepatic lymphocytes, betaretrovirus infection may be present in more PBC patients than we currently can detect solely due to the difficulties in measuring the low levels of the virus. 

### 4.3. Consistency of the Association between HBRV and PBC

The original definition of consistency by Hill [115] was that multiple groups should make the same observations to demonstrate causation over correlation, which was especially important before the advent of molecular methods and data integration [116]. Genetic and molecular biological experiments can provide a molecular mechanism for an observation, relying less on repeated physical observations [116]. In addition, demonstrating a mechanism or phenomenon in various models, including animal, cell culture, or patient samples, can provide further evidence for consistency (and strength) of the association, even if this represents a broader definition than Hill originally proposed. 

Cell culture experiments provide further consistency (Figure 2), where lymph node co-cultivation with primary cholangiocytes results in aberrant PDC-E2 expression, the trigger for AMA production [54]. Mouse models with autoimmune biliary disease have strengthened the link between MMTV and AMA production. As illustrated in Figure 1, models with either inbred or genetic disruption of immunity develop aberrant PDC-E2 autoantigen expression as a result of MMTV infection and produce AMA, providing a cause and effect for linking betaretrovirus infection with AMA production [53]. 

MMTV is involved in breast cancer in mice and HBRV (also referred to as human mammary tumor virus) in humans, as outlined by Lawson and Glenn [89]. They reveal that a human betaretrovirus with 98% similarity to MMTV is detected in 35% to 40% of breast cancers and that HBRV nucleotide sequences are consistently higher in human breast cancer tissue than in normal breast tissue controls. HBRV and MMTV can infect breast cancer cell lines in culture, and the biological mechanism by which MMTV infects and transforms human tissues is similar to what has been determined in mouse models. Essentially, the authors provide multiple sources of data to support the role of HBRV in 40% of human breast cancers [89].

Other researchers have helped to provide evidence for the involvement of HBRV in PBC in case studies and in multicenter randomized controlled trials using ART [60,67]. A report from the UK documented a patient presenting with serum AMA and biopsy confirmed PBC; he had documented HBRV RNA and HIV RNA in serum and made a complete biochemical response with combination ART employing Truvada and lopinavir, an HIV protease inhibitor [60]. 

Ultimately, the activity of ART to reduce betaretrovirus load and improve biochemical and histological parameters of cholangitis in the NOD.c3c4 model [55] and in patients with PBC [55,56,57,58,59,60,66,67] can provide the consistency required for a causal association. Clinical trials using repurposed HIV antiretroviral regimens in PBC patients have consistently shown significant reduction in alkaline phosphatase levels indicative of biochemical improvement in cholangitis [55,56,57,58,59,60,66,67]. A proportion of these studies included liver biopsy assessment and revealed concurrent improvement in cholangitis and the inflammatory activity [57,58,66]. Even if these measures did not meet the stated endpoints of the studies, the clinical improvements coinciding with reduction in HBRV is clear indication that retroviral infection plays a central role in mediating liver disease in PBC [59]. An obvious challenge with these clinical trials includes the use of repurposed HIV ART that was not developed for inhibition of betaretrovirus infection. Various HIV combination ART regimens cause undue side effects and as a result, stringent criteria used for meeting primary endpoints have not been met due to intolerance. Nevertheless, patients maintaining the full dose combination ART developed sustained and clinically meaningful improvement in cholangitis coinciding with reduction in HBRV, as proof of principle of retroviral involvement in the inflammatory disease process [57,59,60].

### 4.4. Biological Gradient/Experiment with Response to Antiviral Therapy

Hill’s original intent for the *biological gradient* was to establish a dose-response curve to relate exposure to a given disease [116]. The biological gradient of diminished viral load may be readily achieved by implementing repurposed HIV ART, which has been demonstrated in vitro against betaretrovirus using several RT inhibitors and the HIV protease inhibitor, lopinavir [102]. The effect of combination ART has also been demonstrated in the NOD.c3c4 mouse model with MMTV cholangitis [55]. Sharon and colleagues found that a combination of RT inhibitors and retroviral protease inhibitors reduced alkaline phosphatase levels, slowed the development of cholangitis, and decreased the amount of MMTV RNA levels in mouse livers [55]. 

The intervention studies also provide an example of a biological gradient in the NOD.c3c4 model and PBC patients treated with escalation from single class antiviral treatment from RT inhibitors alone to combination ART incorporating HIV protease inhibitors. In these studies, more potent antiretroviral intervention was associated with an augmented biochemical response as recorded by further reduction in alkaline phosphatase in both mice and humans using combination regimens incorporating the HIV protease inhibitor lopinavir boosted with ritonavir [55,110]. In successive studies, PBC patients have demonstrated incremental improvements in alkaline phosphatase with one versus two RT inhibitors and subsequently, with RT inhibitors alone compared with combination ART also incorporating HIV protease inhibitors [57,66,67]. 

A biological gradient was also observed with the development of betaretrovirus antiviral resistance variants in both mouse models and patients developing viral breakthrough [55,129]. In these examples, the increased viral load correlated with increased levels of alkaline phosphatase signaling a direct role for viral infection in the development of biochemical cholangitis. 

The moderate success of antiviral therapies in improving PBC biochemical endpoints and reversing the cellular phenotype and disease development is strong evidence that a retrovirus (HBRV) is involved with the disease progression. Now we have established the capacity of combination ART to impact on progressive disease and patient symptoms, the search is now on to identify efficacious and well-tolerated regimens for clinical trials involving PBC patients unresponsive to standard of care [59].

### 4.5. Temporality

Demonstrating temporality is challenging for an infectious process for diseases with an insidious onset, like PBC. Studies such as these require use of archival specimens from population studies to look for evidence of infection prior to the development of disease. For example, the Alberta Tomorrow Project has prospectively collected medical data from over a million individuals in Alberta, Canada and biological samples from more than 55,000 subjects over a period of 14 years [130]. Despite this resource to study viral prevalence in individuals prior to development of disease, it is not feasible to study the temporality of HBRV infection in PBC because of the extremely low incidence of PBC (~1:40,000 patient years), even if we had a more sensitive serological assay [91]. 

However, the study of liver transplant recipients with PBC has provided a framework for studying temporality because disease recurs in over 50% of patients over time; moreover, recurrent PBC more closely models an infectious disease rather than an autoimmune process [28,131]. Disease recurrence is documented by the standard diagnostic features: (i) development of abnormal cholestatic liver tests, (ii) histological appearance of immune destruction of bile ducts, and (iii) serum AMA [28,131]. However, diagnosis of recurrence is not straightforward as the histology can be confused with allograft rejection, and original reports suggested that serum AMA tended to persist in patients with and without recurrence, as does aberrant expression of PDC-E2 on bile ducts in the allograft [132]. With the more recent finding that the prevalence of PBC recurrence tends to increase over time [28], it is reasonable to suggest that given time, the presence of serum AMA and ductular expression of PDC-E2 signal the development of recurrent disease. Either way, the appearance of the disease-specific phenotype in the allograft suggests that the same factors that trigger PBC are still present in recipients following liver transplantation [131]. 

Several lines of evidence point towards infection triggering disease recurrence as opposed to autoimmunity. First, patients on more potent immune suppressive regimens based on the calcineurin inhibitor tacrolimus develop earlier and more severe recurrence compared with those on cyclosporine. Not only is cyclosporine a less-potent immunosuppressant but it also displays broad spectrum antiviral activity against many viruses due to the ability to inhibit cyclophilins required for several key steps in the viral lifecycle [133]. Indeed, cyclosporine demonstrates antiviral activity against betaretroviruses in vitro and can partially inhibit the production MMTV from MM5MT breast cancer cells [102]. It is not just the antiviral activity of some immune suppressants, but the amount of immunosuppression employed in general that is linked with PBC recurrence, such as the use of the additional agent, mycophenolate mofetil [28,51]. A third issue is that early biochemical changes soon after liver transplantation can predict whether PBC will recur in a patient. If a patient develops increased alkaline phosphatase or bilirubin indicative of cholestasis within the first year following liver transplantation, their risk of developing PBC again is doubled [28]. It is understood that several factors can trigger increased cholestatic liver tests soon after liver transplantation and that the histological features may not be sufficiently characteristic to make a distinct diagnosis of say, acute rejection compared with recurrent PBC, as both processes may co-exist. However, demonstration of early cholestasis with eventual recurrent disease in a large international cohort of transplant recipients with PBC demonstrates a clear temporal link, in keeping with an infectious disease process. 

While we have not demonstrated a direct link with HBRV infection and recurrent disease following liver transplantation, we have shown that patients with recurrent PBC may benefit from combination antiretroviral therapy in open label studies [58]. Also, we have argued that recurrent PBC is more in keeping with infectious disease, rather than an autoimmune process [131]. For other infectious diseases following liver transplantation, such as hepatitis C infection, infection occurs soon after liver transplantation with evidence of elevated liver enzymes appearing soon after the development of viremia [134]. The liver biopsy usually shows a mild hepatitis process associated with features of alloimmunity; acute rejection often accompanies inflammatory processes in liver transplant recipients [134]. Although the acute hepatitis settles down, patients then progress to developing chronic liver disease in the absence of antiviral therapy and those receiving potent immunosuppression regimens are more likely to develop more severe disease with recurrent hepatitis C virus infection [134]. As the immunosuppression levels are highest following liver transplantation for all recipients, the temporal basis of recurrent disease favors a infectious disease process as opposed to recurrence of autoimmune disease [131]. 

### 4.6. Plausibility and Coherence

The plausibility and coherence of an association imply that the hypothesis is biologically plausible and is consistent with the understood natural history and biology of the disease. It is not known why autoimmune diseases, in general, are more common in women. Certainly, the reason for the increased incidence of PBC in adult women is unclear but can be explained by the effects of female hormone production on betaretrovirus replication. Both MMTV and HBRV have steroid hormone-responsive elements in the long terminal repeats that maximally promote viral transcription during pregnancy. This results in maximal virus production soon after giving birth to ensure viral transmission via breast milk. These elements can be stimulated by corticosteroids and female hormones, as demonstrated with the increased expression of MMTV in vivo and in mice [135,136]. Accordingly, the stimulation of betaretroviruses by lactogenic hormones may provide an explanation of how hormone replacement therapy and the younger age of first pregnancy can increase a woman’s risk of PBC [26]. The need for the hormone-responsive expression of betaretrovirus may also account for the lack of PBC in children and the exacerbation of PBC during and after pregnancy [137]. 

### 4.7. Analogy 

Numerous viruses exist that cause cancer and inflammatory diseases modulated by genetic predisposition. In mice, we have discussed the comparison with MMTV, which is linked with breast cancer, lymphoma, and renal carcinoma, as well as autoimmune biliary disease, type 1 diabetes, and rheumatoid arthritis [53,89,138]. An example of analogous retrovirus infection in humans include HTLV-1 that leads to the development of adult T-cell leukemia/lymphoma in approximately 5% of infected individuals and other inflammatory/autoimmune diseases like HTLV-associated pulmonary disease and Sjögren’s syndrome in a smaller proportion of patients [120,121]. Hepatitis B virus causes hepatocellular cancer, liver disease, and a series of extrahepatic diseases such as arthritis and glomerulonephritis that predominantly appear in children, and the vasculitides such as polyarteritis nodosa, which is more predominant in Indigenous North American populations [122].

The comparison of hepatitis B virus with HBRV in Indigenous Americans has further repercussions because PBC is far more prevalent and severe in Indigenous Canadians [8,9,10]. The First Nation populations in coastal British Columbia have an estimated ten-fold increase in prevalence of PBC and they exhibit a far greater burden of symptoms compared with Canadians of European descent [8,9,10]. Data from the Canadian Network of Autoimmune Liver disease (CaNAL) report that the Indigenous population were the only ethnic group to have impaired transplant-free and end stage liver disease event-free survival compared with all other ethnic populations in Canada [8]. Approximately 25% of all liver transplantation referrals in British Columbia are from the Indigenous population, whereas they only comprise 3.8% of the demographic population [10]. 

It is not known why PBC is so severe in Indigenous Canadians, but notable that this population also has a higher prevalence of autoimmune diseases such as rheumatoid arthritis, systemic lupus erythematosus, and multiple sclerosis [11]. This population is derived from the recently described Beringians, who migrated from Europe 17,000 to 14,000 years ago and became physically and genetically isolated from Europeans at that juncture. Some of the Indigenous populations of North America appear to have retained a higher frequency of HLA-related risk alleles for PBC that have a diminished mean allelic frequency in Europeans [139]. This raises the question whether the susceptibility to autoimmunity in general is related to genetic predisposition uninfluenced by European habitat, the environmental effects with introduction of infections from Europeans, or a combination of both [11].

## 5. Conclusions

Since the discovery of betaretroviral particles in human breastmilk in the 1970s, the link between a virus and breast cancer has been investigated. More recently, the link of betaretroviruses and autoimmune disorders, including PBC, has been proposed, and a substantial amount of research has been done to support this proposed link. 

Understanding the involvement of a viral trigger in an uncommon disease is not simple. While we have presented evidence for a link between HBRV and the autoimmune phenotype in PBC patients’ samples, biliary epithelium in vitro, and with MMTV in mouse models, there is still much to learn about the viral mechanisms involved in modulating metabolism and altering immune responses [107]. We have also addressed some of the Hill criteria and demonstrated that antiretroviral therapies improve biochemical outcomes of PBC patients and reduce viral load within the liver, both in animal and human studies. While we have not stated endpoints required to prove superiority of a specific treatment strategy, we have demonstrated a proof of principle that combination ART maintains an antiviral effect and is linked with sustained and clinically meaningful reduction in cholestasis [57].

The main limiting factor for investigating HBRV in humans is the low viral load and the lack of a sensitive and reproducible assay. We are currently working on establishing an interferon release assay to achieve this goal [46]. Some research groups have not found evidence using PCR or immunochemistry because their methods were not sensitive enough, or they were not looking at the site of maximal infection within perihepatic lymph nodes. Another concern raised in previous studies has been contamination, which has been addressed through the use of gold standard assays to detect viral integrations using a stringent pipeline that excludes interference with endogenous viral sequences [69].

Here, we make the case that HBRV is associated with PBC in humans. At this time, no evidence exists to contradict the involvement of a viral trigger, except for the failure by some groups to detect evidence of the virus in liver samples. Negative studies do not necessarily mean that a theory is incorrect, especially when diagnostic tests do not have sufficient sensitivity. Concerns raised in various negative studies have been addressed, and the development of more sensitive assays for the virus will certainly resolve any remaining concerns. Other theories for a bacterial or xenobiotic trigger for PBC have little experimental support and would not satisfy the Hill criteria as described here.

While work to date has not resulted in identification of a well-tolerated regimen, ART repurposed from HIV treatments has demonstrated sufficient activity to pursue this line of research for PBC patients because current treatments do not impact on symptoms. With the evidence that PBC may be modulated by ART, newly described, and better tolerated antivirals can be evaluated to assess whether specific regimens can provide better outcomes for treatment and prevention of this disease.

## Figures and Tables

**Figure 1 viruses-14-01941-f001:**
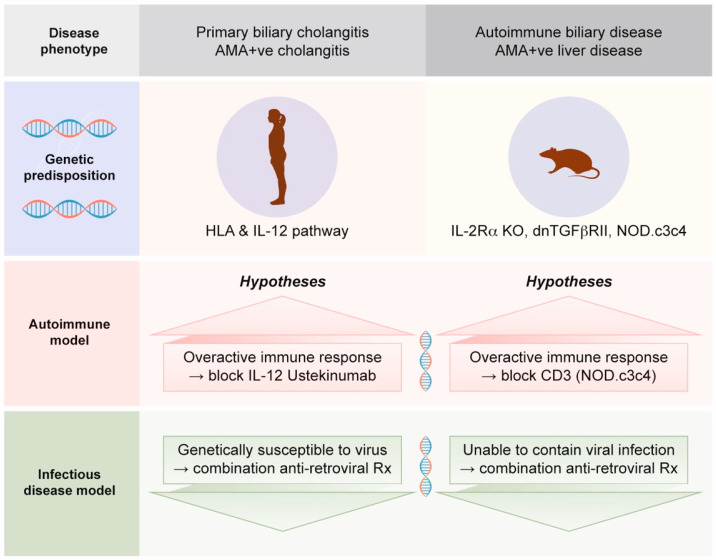
Autoimmune and infectious disease models for interpreting the genetic predisposition to biliary disease. Genome-wide association studies of patients with PBC and animal models of spontaneous autoimmune biliary disease provide alternative hypotheses for the development of cholangitis. In the autoimmune model, it is assumed that the immune system is the origin of disease; therefore, treatment is geared towards immunosuppression or immune targeted therapy. Whereas if cholangitis is triggered by a betaretrovirus, the genetic susceptibility is likely to provide a degree of immunodeficiency to lower resistance to viral infection, and therefore anti-retroviral therapy is the treatment of choice.

**Figure 2 viruses-14-01941-f002:**
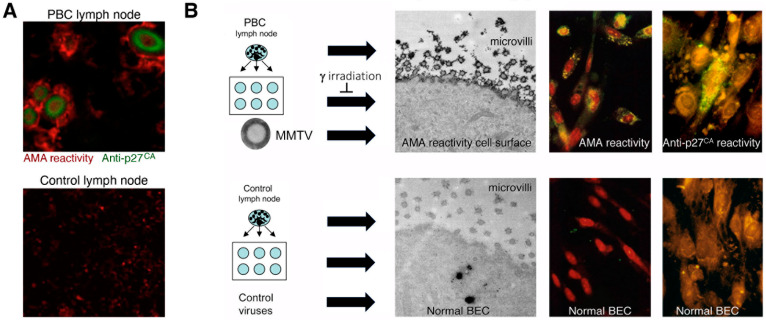
(**A**) Perihepatic lymph nodes from patients with PBC are infected with HBRV and have evidence of HBRV anti-p27 Capsid reactivity (green label) is detected in cells that express aberrant AMA reactivity (red label), whereas the control lymph node only shows AMA reactivity in mitochondria and no evidence of HBRV infection. (**B**) Co-cultivation studies show that biliary epithelial cells (BEC) develop mitochondrial autoantigen expression with AMA reactivity when incubated with (i) PBC patient’s lymph node homogenates, (ii) conditioned supernatants, and (iii) MMTV. Perihepatic lymph nodes from PBC and control patients were homogenized and co-cultured with normal BEC isolated from liver transplant recipients. Subsequently, the PBC lymph node-conditioned BEC demonstrated AMA reactivity as observed by immunoelectron microscopy and evidence of HBRV anti-p27 Capsid proteins by immunofluorescence. Conditioned supernatants had the same effect, whereby only the PBC-conditioned supernatants triggered AMA reactivity. This process was abrogated by γ-irradiation. Normal BEC incubated with supernatant from MMTV-producing MM5MT cells also showed AMA reactivity and anti-p27 Capsid immunofluorescence, whereas control viruses had no such effect. Copyright 2003 National Academy of Sciences.

**Figure 3 viruses-14-01941-f003:**
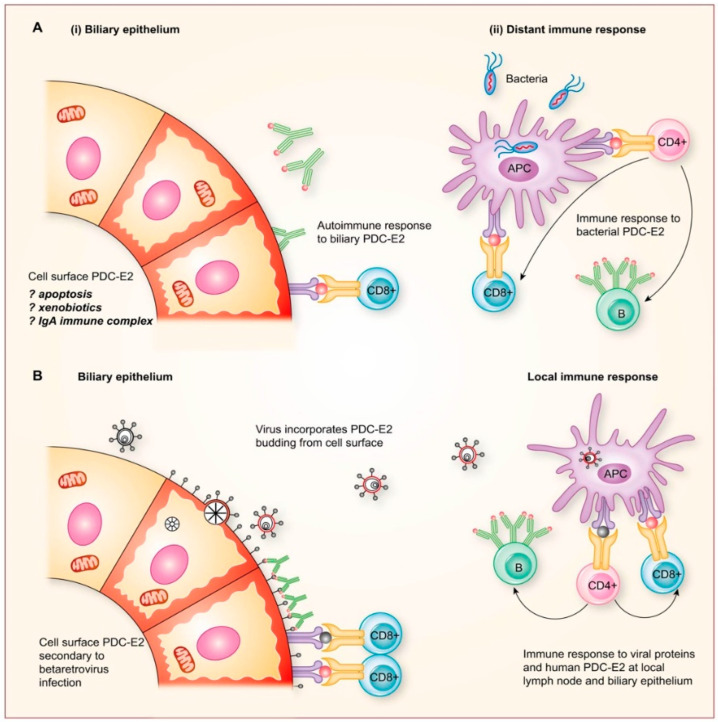
(**A**) Model for generation of autoimmune responses to PDC-E2 with bacterial infection: (i) in a two-hit process, biliary epithelial cells present PDC-E2-like proteins on the biliary epithelium; and (ii) bacterial infection results in PDC-E2 presentation on antigen-presenting cells (APC) breaking tolerance to human PDC-E2 [arrows represent T cell help]. The subsequent autoimmune response homes in on the aberrant expression of PDC-E2 in biliary epithelium [? represents factors that may precipitate cell surface PDC-E2]. (**B**) In a single-hit process, human betaretrovirus infects biliary epithelium leading to PDC-E2 expression on the cell surface. The virus either incorporates PDC-E2 while budding from the cell surface or exits with PDC-E2 in exosomes (not shown). APCs then present PDC-E2 and viral proteins, resulting in a bystander immune response to PDC-E2 and an immune response to viral and self-proteins expressed on biliary epithelium (Adapted from Wasilenko et al. [95] with permission).

**Figure 4 viruses-14-01941-f004:**
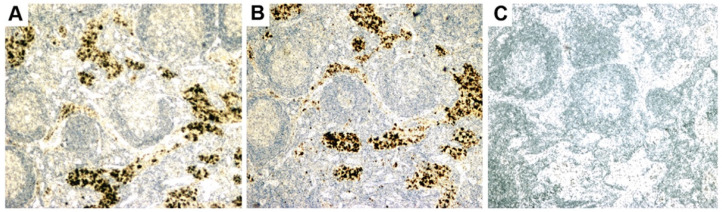
HBRV Capsid and Surface proteins in PBC perihepatic lymph nodes. These studies show immunostaining with (**A**) anti-MMTV p27 Capsid antibody and (**B**) anti-MMTV gp52 Surface antibody in consecutive layers of PBC lymph node showing betaretrovirus proteins in the same distribution. In contrast (**C**), the control lymph node has no anti-MMTV p27 capsid antibody staining (H&E 200× magnification). Reproduced from *Journal of Hepatology* with permission, reference [71].

**Figure 5 viruses-14-01941-f005:**
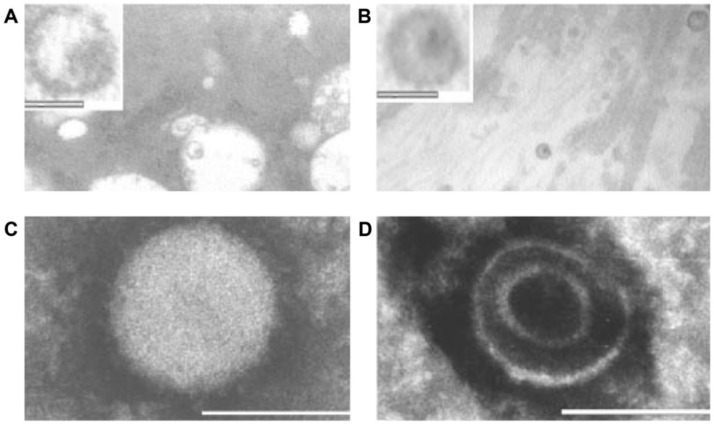
Electron microscopy studies reveal virus-like particles ex vivo and in vitro in samples from patients with PBC. (**A**,**B**) Freshly isolated biliary epithelial cells from a PBC patient showing virus-like particles in the extracellular space with a distinct envelope and electron-dense core (Inset shows particle at 5× magnification; white bar represents 100 nm). (**C**,**D**) Similar particles ranging in size from 110 to 120 nm and showing features consistent with B-type particles were observed in negatively stained pellets from PBC-conditioned media in BEC supernatants. The negative stain has penetrated the structure in panel D to reveal the nucleoprotein core. Copyright 2003 National Academy of Sciences.

## Data Availability

HBRV proviral integration data located at NCI Retroviral Integration Database; rid.ncifcrf.gov [90].

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
