# Peer review of "Linking Human Betaretrovirus with Autoimmunity and Liver Disease in Patients with Primary Biliary Cholangitis"

_viruses, 2022, doi:10.3390/v14091941_

Round 1

Reviewer 1 Report

This review by Syed et al. provides a discussion of the relatively rare autoimmune disease, primary biliary cholangitis (PBC).  The authors provide information about the diagnosis of PBC, particularly its association with anti-mitochondrial antibodies (AMA).  PBC, like many autoimmune diseases, is more prevalent in women.  The major focus of this review is to argue that a mouse mammary tumor virus (MMTV)-like agent (called human betaretrovirus or HBRV) is the causal agent of PBC.  A similar argument has been made for the role of MMTV in breast cancer for many years (human mammary tumor virus or HMTV), yet confirmatory evidence has not been provided for either HBRV or HMTV.  The lack of rigorous molecular and virological experiments by this laboratory and their failure to be reproduced in other laboratories limits enthusiasm for this manuscript.

Major comments:

1.    If HBRV is an infectious agent, it should be possible for more laboratories to reproduce these findings.

2.    Line 169 – MMTV is not expressed at high levels in the liver.  Tropism for retroviruses has been shown to be determined by receptor binding specificity and/or by transcription factor binding sites within the LTR.  Without evidence for changes within these regions relative to MMTV, it is unlikely that HBRV could have liver tropism.

3.    The figures within the review are almost entirely from the senior author’s laboratory.  A review should provide a more unbiased view. 

4.    Line 717 – The authors argue that PBC is more common in women due to the hormone responsive elements in the HBRV LTR.  However, autoimmune diseases are more frequent in women, even in diseases not associated with HBRV.

5.    Line 783 – The ability to show HBRV integration sites is not a definitive argument since they could also result from contamination.

6.    Despite published reports of HBRV association with PBC for at least 20 years, most clinical reviews of PBC have not mentioned a retroviral etiology or use of anti-retroviral drugs.    

Minor comments:

1.    Line 174 – Like other retroviruses, MMTV makes many RT errors.

2.    Line 182 – Exogenous MMTV levels in lymphoid cells are quite low compared to lactating mammary glands.

3.    Line 192 – Superantigen presentation leads to the deletion of specific T-cell subsets.

4.    Multiple places – “Envelop” should be “envelope”.

Reviewer 2 Report

This is a clearly written review. The sum of evidence shows that HBRV is associated with PBC in humans. The authors are correct to point out that the failure by some groups to detect evidence of the virus in liver samples does not necessarily rule out a role for HBRV in PBC. This reviewer agrees that development of more sensitive assays for the virus will likely resolve any remaining controversies.

Author Response

We thank the reviewer and agree with their comments and suggestions.

Reviewer 3 Report

The paper focuses on the relationships between HBRV and PBC. This topic has a special relevance for oncology too, in that HBRV is linked also to human neoplasms, breast cancer in particular. 

Authors during the years have contributed many relevant data that indicate HBRV as etiological agent of PBC.

This paper analyzes in detail the etiopathogenesis of PBC enlightening the links between HBRV, immune system, and autoimmunity.

Moreover and advisably, authors decided to analyze all the available data in the light of Koch’s postulates and Hill’s criteria, adding a table “Hill’s criteria applied to HBRV infection and PBC”.

The table lists five criteria, whereas the text analyzes also “temporality” and “analogy”. Could authors explain this difference? In case, thy could complete the table.

Author Response

We thank the reviewer for their kind comments. We have amended the table: Hill’s criteria applied to HBRV infection and PBC.

We have added the two additional criteria, “temporality” and “analogy” as suggested.